# Molecular Dynamics Study on Wear Resistance of High Entropy Alloy Coatings Considering the Effect of Temperature

**DOI:** 10.3390/ma17163911

**Published:** 2024-08-07

**Authors:** Xianhe Zhang, Zhenrong Yang, Yong Deng

**Affiliations:** 1Hebei Key Laboratory of Mechanics of Intelligent Materials and Structures, Shijiazhuang Tiedao University, Shijiazhuang 050043, China; 2Hebei Research Center of the Basic Discipline Engineering Mechanics, Shijiazhuang Tiedao University, Shijiazhuang 050043, China; zhenrong.yang@soyotec.com; 3School of Civil Aviation, Northwestern Polytechnical University, Xi’an 710012, China; 4Research & Development Institute of Northwestern Polytechnical University in Shenzhen, Shenzhen 518063, China

**Keywords:** high entropy alloy coating, wear resistance, temperature, dislocation density, lattice disorder

## Abstract

High entropy alloys have excellent wear resistance, so they have great application prospects in the fields of wear resistance and surface protection. In this study, the wear resistance of the FeNiCrCoCu high entropy alloy coating was systematically analyzed by the molecular dynamics method. FeNiCrCoCu high entropy alloy was used as a coating material to adhere to the surface of a Cu matrix. The friction and nanoindentation simulation of this coating material were carried out by controlling the ambient temperature. The influence of temperature on its friction properties was analyzed on five aspects: lattice structure, dislocation evolution, friction coefficient, hardness, and elastic modulus. The results show that with the increase of temperature, the disorder of the lattice structure increases, which leads to an increase of the tangential force and friction coefficient in the friction process. At 300 K and 600 K, the ordered lattice structure of the high entropy alloy coating material is basically the same, and thus its hardness is basically the same. However, the dislocation density at 600 K is significantly reduced compared with that at 300 K, resulting in an increase of the elastic modulus of the material from 173 GPa to 219 GPa. At temperatures of 900 K and 1200 K, lattice disorder takes place rapidly, and dislocation density also decreases significantly, resulting in a significant decrease in the hardness and elastic modulus of the material. When the temperature reaches 900 K, the wear resistance of the FeNiCrCoCu high entropy alloy coating decreases sharply. This work is of great value in the analysis of wear resistance of high entropy alloys at high temperature.

## 1. Introduction

The appearance of high entropy alloys breaks the traditional design concept of alloys based on mixing enthalpy and opens up a broad space for composition design for the research and development of new materials. The comprehensive effect of the mixing of various main elements endows the high entropy alloy with high entropy effect in thermodynamics, lattice distortion effect in structure, hysteresis diffusion effect in dynamics, and “cocktail” effect in performance [1,2]. The principle of multi-principal element design based on these four effects results in the high entropy alloy showing many unique properties beyond the traditional alloy. These characteristics cover excellent corrosion resistance [3], wear resistance [4,5,6,7,8,9], excellent fatigue strength, and yield strength under various temperature conditions [10,11], high hardness [12], excellent microstructure stability and thermal stability [13,14], and excellent high-temperature oxidation resistance [15]. In view of their outstanding mechanical and functional characteristics, high entropy alloys show broad application prospects in many fields, such as marine engineering, the automotive industry, energy development, the chemical industry, and biomedicine [16,17,18,19].

FeNiCrCo-X high entropy alloys have become a research hotspot for the relevant scholars in recent years because of their unique physical properties such as high temperature stability [13], high hardness, and high strength [4,5,6,7,8,9,12]. Among them, FeNiCrCoCu high entropy alloy has excellent wear resistance, which is regarded as an ideal high-temperature wear-resistant material, attracting a large number of scholars to study such alloys. At present, it has been proven that this alloy can be achieved and is suitable for use as a thin film application [20,21,22]. Due to the expensive preparation of this high entropy alloy material, it is more reasonable to add it as a coating material to traditional metal surfaces in future practical applications. A new Al_0.75_CrFeNi eutectic high entropy alloy was prepared composed of a disordered body centered cubic phase and an ordered body centered cubic (BCC) phase, showing excellent mechanical properties and high yield strength [23]. Mohsen et al. [24] systematically studied the wear resistance of the Al_x_Co_1.5_CrFeNi_l.5_Ti_y_ system and found that the wear resistance of Co_l.5_CrFeNi_l.5_Ti and A1_0.2_Co_1.5_CrFeNi_l.5_Ti alloys was at least twice that of conventional wear-resistant steel. Braic et al. [25] used (TiZrNbHfTa)N high entropy alloy as a protective coating in the biomedical field to improve the indicators of titanium alloy implants; they pointed out that the hardness of the alloy increased to 30 GPa, and the surface wear rate was lower than 0.2 × 10^−6^ mm^3^ N^−1^ m^−1^. Li et al. [26] found that FeNiCrCoCu high entropy alloy coating on the surface of a Cu matrix can effectively improve the wear resistance of the material by the molecular dynamics method. They reached the conclusion that an increase of the friction velocity results in the reduction of the tangential force affected by material thermal softening and the increase of the normal force affected by material strain rate hardening, and therefore leads to the reduction of the friction coefficient [26]. Yang et al. [27] found that the hardness and Young’s modulus of AlCoCrFe high entropy alloy coatings are about 10 times that of the substrate metal Al. In the interface area between the high entropy alloy coating and the Al substrate, due to the laser heating during the manufacturing process of the high entropy alloy coating, a part of the substrate melts and reacts with other elements in the coating forming several BCC structures which can effectively protect the matrix material from wear. [27]. Doan et al. [8], using AlCoCrFeNi high entropy alloy as a coating material adhered to the surface of a Ni substrate, found that Ni based alloys are protected by high entropy alloy coatings with less wear. With the increase of friction depth, the resistance of tool movement is greater, and the friction coefficient increases. Instead, due to the effects of thermal softening and strain rate hardening, the friction coefficient decreases with the increase of friction speed [8]. Bui et al. [28] investigated the effects of composition, grain size, and cutting depth on the deformation behavior and damage of NiCoCrFe high entropy alloys. It was found that the dislocations increased with the increase of Co content, the Ni_25_Co_25_Cr_25_Fe_25_ high entropy alloy had the highest number of worn atoms, while the Ni_25_Co_25_Cr_30_Fe_20_ high entropy alloy had the lowest number of worn atoms. 

It can be concluded that researchers have mainly focused on studying the room temperature friction performance of high entropy alloy materials and have not studied the overall wear resistance at different temperatures. However, at present, researchers are mainly focusing on the study of the properties of the materials themselves and are not studying their overall wear resistance as coating materials. Also, it is difficult to observe the microscopic deformation mechanism during the experimental process, and effective conclusions cannot be drawn on the impact mechanism of wear resistance at different temperatures. Therefore, in response to the shortcomings of current research for high entropy alloy coatings, this study aimed to analyze the influencing mechanism of wear resistance of FeNiCrCoCu high entropy alloy coatings on Cu matrix at different temperatures of the microstructure by molecular dynamics methods, to provide a theoretical basis for practical applications.

## 2. Materials and Methods

### 2.1. Friction Simulation

In this work, single crystal Cu was used as the matrix material and FeNiCrCoCu high entropy alloy as the coating material. The lattice types of both materials are face centered cubic (FCC). The lattice constant of Cu was set as 3.56 Å; the lattice constant of FeNiCrCoCu high entropy alloy was also set as 3.56 Å. The dimensions of the Cu matrix were 72 Å × 185.2 Å × 52 Å, and the total number of atoms was 62,423. The ambient temperature was set at 300 K, 600 K, 900 K, and 1200 K. The high entropy alloy coating was 21 Å. The same crystal orientations of matrix and coating material in the X axis, Y axis, and Z axis were set as [100], [010], and [001] directions, respectively. The virtual indenter of Lammps was used in this simulation. This nano indentation simulation method has already been widely used [29,30]. The form of force field between virtual indenter and atoms is shown in Equation (1) as follows [31,32,33,34,35,36,37]: (1)$$F(r)=\left\{\begin{array}{cc} -K{\left(R-r\right)}^{2} & r>R\\ 0 & r<R \end{array}\right.$$
where *K* is the specified force constant; *r* is the distance from the atom to the center of the indenter; *R* is the indenter radius. In this simulation, the value of constant *K* is set as 3ev/Å, which is considered to be very reliable in the simulation of multiple nanoindentations [26,29].

After the initial model was built, in order to eliminate the unreasonable structure in this model, the conjugate gradient algorithm [38] was first used to minimize the energy of the model. Then, the model was relaxed. In the relaxation process, periodic boundary conditions were used in the X, Y, and Z directions of the model, and the model was initialized at 300 K. In the next four steps, the Nosé–Hoover hot bath method [39] was used to control the system. We performed time integration on the Nosé–Hoover style non-Hamiltonian equations of motion which are designed to generate positions and velocities sampled from the canonical, isothermal–isobaric, ensembles. The canonical ensembles are a collection of systems with the same number of particles and volume, and the temperature is fixed, but the energy is allowed to fluctuate (NVT). The isothermal–isobaric ensembles are systems with a fixed number of particles, fixed pressure, and fixed temperature (NPT). In the first step, the NPT ensemble [40,41] was used to raise the temperature of the model from 300 K to 1000 K, taking 200 ps. In the second step, the NVT ensemble [41] was used to maintain the model at a temperature of 1000 K for 100 ps. The third step was to use the NPT ensemble to reduce the temperature from 1000 K to 300 K, which takes 200 ps. The fourth step was to use the NVT ensemble model to maintain 100 ps at 300 K. Figure 1a shows the final distribution of the five elements Fe, Ni, Cr, CO, and Cu in the high entropy alloy coating after the model reaches the equilibrium state. 

After the relaxation model reaches the equilibrium state, the atoms in the range of 0~5 Å, 5~10 Å, and 10~52 Å in the Z direction were set as the fixed layer, thermostatic layer, Cu layer as shown in Figure 1b. The high entropy alloy coating was set as the high entropy alloy layer. The atoms in the Cu layer and high entropy alloy layer were Newtonian layers with free moving atoms. The fixed layer was placed at the bottom of the model to avoid movement in space. The thermostat layer consisted of atoms adjacent to the fixed layer to maintain the temperature of the system constant. The simulation only works on the thermostat and Newtonian layers. Periodic boundary conditions were used in the X and Y directions of the model, and contractive boundary conditions were used in the Z direction. The atoms in the boundary layer were fixed as rigid bodies, and the Cu layer and high entropy alloy layer were set as the NVE ensemble. The constant temperature layer was set as the NVT ensemble, and the temperature was controlled at 300 K. The initial position of the indenter was located above the matrix and pressed into the matrix at a speed of 10 m/s along the negative direction of the Z axis, with a depth of 10 Å, and then rubbed along the positive direction of the Y axis at a speed of 10 m/s, as shown in Figure 1b.

### 2.2. Nanoindentation Simulation

In the nanoindentation model, single crystal Cu was used as the matrix material and FeNiCrCoCu high entropy alloy was used as the coating material. The same lattice types of both materials were set. The model size of the Cu matrix was 100 Å × 100 Å × 55 Å, and the total number of atoms was 47,111. The ambient temperature was also set at 300 K, 600 K, 900 K, and 1200 K. The high entropy alloy coating was also set as 21 Å. The substrate and coating adopted the same crystal orientation as above. In order to eliminate unreasonable structure in the model, the conjugate gradient algorithm [38] was used to minimize the energy, and then the model was relaxed. The relaxation process is consistent with the friction model above. The final distribution of the five elements Fe, Ni, Cr, Co, and Cu in the high entropy alloy coating reached the equilibrium state after the relaxation process.

After the model reached the equilibrium state, the atoms in the range of 0~5 Å, 5~10 Å, and 10~55 Å in the Z direction were set as the boundary layer, constant temperature layer, and Cu layer, and the high entropy alloy coating was set as the high entropy alloy layer, as shown in Figure 2. Periodic boundary conditions were used in the X and Y directions of the model, and contractive boundary conditions were used in the Z direction. The atoms in the boundary layer were fixed as rigid bodies, and the Cu layer and high entropy alloy layer were set as the NVE ensemble. The setting of the constant temperature layer adopted the Langevin method to control the temperature. The temperature was controlled as 300 K, 600 K, 900 K, and 1200 K, respectively. The bottom of the virtual indenter contacted with the top of the simulation model, and there was no pressure. After the simulation position was determined, it started to press into the matrix material at the speed of 10 m/s. The loading process lasted for 300 ps, and the pressing depth was 30 Å. After that, the unloading started at the same speed as the loading, and the unloading time lasted for 300 ps. 

## 3. Results

### 3.1. Force Displacement Curve

Four temperatures were designed for friction simulation. The change of tangential force during friction is shown in Figure 3a, and the change of normal force is shown in Figure 3b. With the increase of temperature, the tangential force has a slow increasing trend, and the variation range is small. The general difference between 300 K and 1200 K is about 20 nN. After the temperature rises, the normal force significantly decreases. When the temperature increases from 300 K to 900 K, the normal force decreases slowly; the change value of the normal force decreases by about 20 nN. When the temperature reaches 1200 K, the change of the normal force amplitude increases, and the value of the positive force decreases by about 40 nN from that found below 900 K. It can be seen from the force curve that the FeNiCrCoCu high entropy alloy has excellent high temperature resistance at 300–900 K.

The virtual indenter was also used to simulate nanoindentation. The force displacement curve in the unloading process was linearly fitted to the initial unloading position. The hardness and elastic modulus of the material were calculated using the Oliver Pharr method [42,43]. During the virtual indenter pressing process, the pressure generated on the model is shown in Figure 4. The material went through elastic deformation and plastic deformation at different temperatures, which was similar to the loading curve of traditional alloys. At 300 K and 600 K, the slope of the curve in the elastic stage is basically the same as that in the unloading process. At 900 K and 1200 K, the slope of the curve in the loading stage is higher than that in the unloading stage, indicating that the friction properties of the material have changed permanently in the friction process.

### 3.2. Effect of Temperature on Lattice Structure

Figure 5 shows the evolution of the lattice structure in the friction process at different temperatures. It shows that the number of hexagonal close-packed (HCP) atoms increases after friction with the increase of temperature. The corresponding defect structures indicate new stacking faults were induced by friction, consistent with the previous observation that the planar slip mainly governs the deformation [44,45]. At a temperature of 300 K, a small amount of stacking faults appeared at the position of the high entropy alloy coating during the friction process. The overall structure was very stable, consisting almost entirely of FCC crystal phase, and a little disordered structure appeared on the coating surface. At a temperature of 600 K, the coating material basically did not produce stacking faults after friction, but there was a small amount of stacking faults in the matrix material, and the overall structure was still very stable. When the temperature reached 900 K, stacking faults across the interface between the two materials appeared, and disordered structures increased in the Cu matrix. At a temperature of 1200 K, the lattice structure was still mostly FCC phase in the high entropy alloy coating, and only a little part of the disordered structure was increased. The stability of the lattice structure in the Cu matrix decreased greatly. During the friction process, a large number of disordered structures appeared and began to melt, and the stacking faults also began to decrease at 1200 K. Due to the softening of the lattice with increasing temperature, the deformation mechanism transformed from stacking faults to lattice disorder.

Figure 6 shows the simulation process of nanoindentation under different temperatures. The lattice structure changed during the loading process of the indenter. At a temperature of 300 K, a small amount of stacking faults appeared in the high entropy alloy coating and Cu substrate during the indentation process. The overall structure was very stable, formed by FCC crystal phase, and a little disordered structure appeared on the coating surface. At a temperature of 600 K, there was basically no stacking faults in the coating material after friction, a small amount of stacking faults appeared in the Cu matrix material, but the overall structure was still very stable. When the temperature reached 900 K, stacking faults across the interface between the two materials appeared, and disordered structures increased in the Cu matrix. At a temperature of 1200 K, there was mainly FCC crystal phase in the high entropy alloy coating, and the disordered structure only increased slightly. The stability of the lattice structure in the Cu matrix became worse. During the pressing process, a large number of disordered structures appeared, the Cu matrix began to melt, and the stacking fault also began to decrease.

### 3.3. Effect of Temperature on Lattice Structure

Figure 7 shows the evolution of dislocation at four different temperatures in the friction process. At a temperature of 300 K, there were mainly Shockley incomplete dislocations at the initial indentation position and the middle region of the indenter in the friction process. There was also a small number of stair rod incomplete dislocations generated, connecting Shockley incomplete dislocations between different stacking faults. At an ambient temperature of 600 K, the Shockley incomplete dislocation was mainly located at the initial indentation position of the indenter and the middle region in the friction process. Dislocation slip phenomenon began to appear in the friction process, and there was a reaction in the Shockley incomplete dislocation slip process in the two regions. At a temperature of 900 K, the Shockley incomplete dislocation appeared as a large range along the Y-axis direction from the initial pressing position and the middle region of the friction process, and the Shockley incomplete dislocation between different stacking faults was connected by the stair rod incomplete dislocation, which damaged the elastic recovery ability of the material. At a temperature of 1200 K, the Shockley incomplete dislocation in the high entropy alloy coating and the Cu matrix showed obvious delamination during the friction process, and a small amount of complete dislocation appeared on the contact surface of the two materials. The Shockley incomplete dislocation in the high entropy alloy coating was a form of line defects continuity. However, there was a uniform dense distribution of short dislocations, and no longer a long dislocation line in the Cu matrix. This might be due to a large number of surface defects in the Cu matrix under the high temperature environment, which hindered the generation of line defect dislocation.

Figure 8 shows the evolution of dislocation at four temperatures in the nanoindentation process. After the indentation depth of 10 Å, the dislocations mainly appeared in one side at temperatures of 300 K, 600 K, and 900 K. The length of the dislocation line was the smallest, and the deformation recovery ability of the material was enhanced at a temperature of 600 K. During the loading process of the indenter, the Shockley incomplete dislocation appeared as a sliding phenomenon, moving from one side to the other side, and finally existed in the whole matrix at the end of the loading. At a temperature of 1200 K, the Shockley incomplete dislocations in the high entropy alloy coating and the Cu matrix showed an obvious delamination phenomenon during the loading process. When the indentation depth was 20 Å, the Shockley incomplete dislocations were continuous line defects in the high entropy alloy coating. However, in the Cu matrix, there were uniformly distributed short dislocations, and no longer dislocation lines. When the indentation depth was 30 Å, there was a small number of segment dislocations in the matrix material. This might be due to the damage of the contact surface caused by the penetration of the coating into the matrix under the pressure of the indenter at high temperature. This damage could have resulted in a large number of surface defects in the matrix, which hindered the generation of dislocations.

## 4. Discussion

### 4.1. Effect of Temperature on Morphology and Friction Coefficient

Figure 9 shows the atomic wear morphology on the material surface after friction simulation at four temperatures. With the increase of temperature, it can be seen that the wear degree of coating obviously increases.

The friction coefficients of FeNiCrCoCu high entropy alloy coating material at 300 K, 600 K, 900 K, and 1200 K were obtained by using Amontons Coulomb friction law [46,47]. As shown in Figure 10, the effect of temperature on the friction coefficient is analyzed. With the increase of temperature, the friction coefficients gradually increased, and the fluctuation range also continued to increase. When the temperature rose from 900 K to 1200 K, there was a large increase, and the fluctuation was more intense. This was because the temperature was close to the melting point of the Cu matrix material, resulting in a significant decrease in the hardness of the material and a significant decrease in the value of the positive force. However, the high entropy alloy material was relatively stable at this temperature. The friction depth of the indenter was less than the thickness of the high entropy alloy coating, and the value of the tangential force in the friction process was basically stable, resulting in a significant increase in the friction coefficient of the material. The friction coefficient data were also basically consistent with 0.5~0.75 measured by Li et al. in the experiments [48].

### 4.2. Effect of Temperature on Elastic Modulus and Hardness

The change of ambient temperature would have a great impact on the hardness and elastic modulus of the material. In Figure 4, the change of the positive force under different temperature environments can be seen intuitively. The Oliver Pharr method [42,43] was used to calculate hardness and elastic modulus. The calculation expression of the O-P method is as follows [42,43]:(2)$$h_{c}=h-\varepsilon \frac{P(h)}{S}$$
where *h*_c_ is the contact depth; *h* is the maximum indentation depth; *P* is the corresponding load of *h*; *S* is the slope of the unloading curve; *ε* is the correction coefficient. Tan used the O-P method and nanoindentation technology to carry out an experiment, and the result was reliable when *ε* was 0.75 [49]:(3)$$A\left(h_{c}\right)=24.5h_{c}^{2}+C_{1}h_{c}^{1}+C_{2}h_{c}^{1/2}+C_{3}h_{c}^{1/4}+\ldots +C_{8}h_{c}^{1/128}$$
(4)$$H=\frac{P}{A(h_{c})}$$
(5)$$E_{r}=\frac{S\sqrt{\pi }}{2\sqrt{A(h_{c})}}$$
(6)$$\frac{1}{E_{r}}=\frac{\left(1-\nu^{2}\right)}{E}+\frac{\left(1-\nu_{i}^{2}\right)}{E_{i}}$$
where *C*_1_, *C*_2_, … *C*_8_ are the correction constants; *H* is Rockwell hardness; *E*_r_ is the reduced elastic modulus; *v, v_i_* are the Poisson’s ratio of matrix material and indenter, respectively; *E, E_i_* are the elastic modulus of coating material and indenter, respectively. Since the virtual indenter used in this paper is loaded, the elastic modulus can be taken as infinity, so Equation (6) could be simplified as follows:(7)$$\frac{1}{E_{r}}=\frac{\left(1-\nu^{2}\right)}{E}$$

According to the loading curve in Figure 4 and the fitting curve in the unloading process, the Poisson’s ratio of similar materials was about 0.32 [50], and the data shown in Table 1 could be obtained. The hardness and elastic modulus corresponding to each model obtained after calculation was as shown in Figure 11.

As shown in Figure 11, the hardnesses of the FeNiCrCoCu high entropy alloy coating are basically the same at 300 K and 600 K. When the temperature was raised to 900 K, there was a large decrease from 17.4 GPa to 10.2 GPa. When the temperature was raised from 900 K to 1200 K, the hardness decreased slightly, from 10.2 GPa to 8.6 GPa. When the temperature increased from 300 K to 600 K, the elastic modulus increased significantly, from 173 GPa to 219 GPa. As the temperature continued to rise, the elastic modulus began to decline rapidly, which was also consistent with the slope change during the unloading pressure process. At low temperature, the high entropy alloy coating material had higher hardness, elastic modulus, and better wear resistance. In the high temperature environment, the hardness decreased significantly, and the friction performance of the material also decreased continuously, which is easier to be destroyed in the friction process. Luo et al. [51] obtained an elastic modulus of 201 GPa and a hardness of 17.2 GPa of FeNiCrCoCu high entropy alloy at 293 K, which is very close to the simulation results at 300 K in our work. Additionally, the results are also basically consistent with the results obtained by Deng et al. [52] in their experiment. Through the above results in Figure 6, it could be found that the degree of atomic disorder increases with temperature when the temperature reaches 600 K, which leads to decreasing hardness. Based on the results of Figure 8, the density of dislocations at 600 K is significantly reduced compared with that at 300 K, resulting in an increase of the elastic modulus of the material from 173 GPa to 219 GPa. At temperatures of 900 K and 1200 K, the densities of dislocations increase rapidly resulting in a significant decrease in the hardness and elastic modulus of the material.

## 5. Conclusions

In this study, the wear resistances of the FeNiCrCoCu high entropy alloy coating material were comprehensively analyzed by the molecular dynamics method. The effects of temperature on the evolution of lattice structure, dislocation, friction coefficient, and hardness of the high entropy alloy coating material were analyzed. Through the above research and analysis, the following conclusions were obtained:(1)In the friction process of the FeNiCrCoCu high entropy alloy coating, the value of the normal force decreases greatly due to thermal softening with the increase of temperature. The friction coefficients increased with the temperature, especially when the temperature reached 1200 K. This is because the high temperature leads to the increase in the proportion of disordered atoms in the material. At 300 K, 600 K, and 900 K, the ordered lattice structures of the high entropy alloy coating material are basically the same. At a temperature of 1200 K, the proportion of disordered lattice structures increases rapidly, resulting in a significant decrease in the normal force, which caused a significant decrease in the friction coefficient.(2)When the temperature was increased from 300 K to 600 K, the elastic modulus increased significantly. The density of dislocations at 600 K was significantly reduced compared with that at 300 K, resulting in the increase of the elastic modulus of the material from 173 GPa to 219 GPa. At temperatures of 900 K and 1200 K, the densities of dislocations increased rapidly, resulting in a significant decrease in the elastic modulus of the material. The hardnesses of the FeNiCrCoCu high entropy alloy coating were basically the same at 300 K and 600 K. Additionally, the hardness kept decreasing with increasing temperature, when the temperature reached 600 K. It was found that the degree of atomic disorder increased with temperature when the temperature reached 600 K, which lead to decreasing hardness.

There are still many shortcomings in this article. We did not analyze the effect of changes in the proportion of elements in the FeNiCrCoCu high entropy alloy material on wear resistance. Subsequent analysis can be conducted by changing the proportion of elements to obtain the external influencing mechanism. Also, we did not study the roughness of the material surface. In the future, the influence of roughness on the wear resistance of materials can be studied by changing the surface morphology of the material.

## Figures and Tables

**Figure 1 materials-17-03911-f001:**
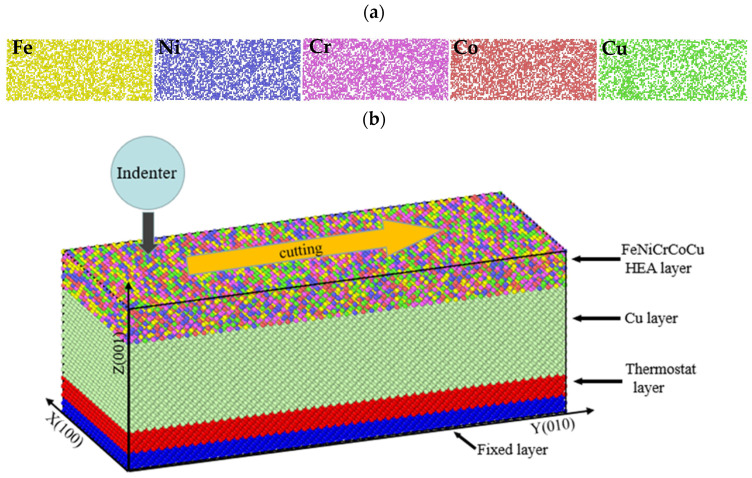
(**a**) Distribution of various atoms in equilibrium state, and (**b**) schematic diagram of friction model area distribution.

**Figure 2 materials-17-03911-f002:**
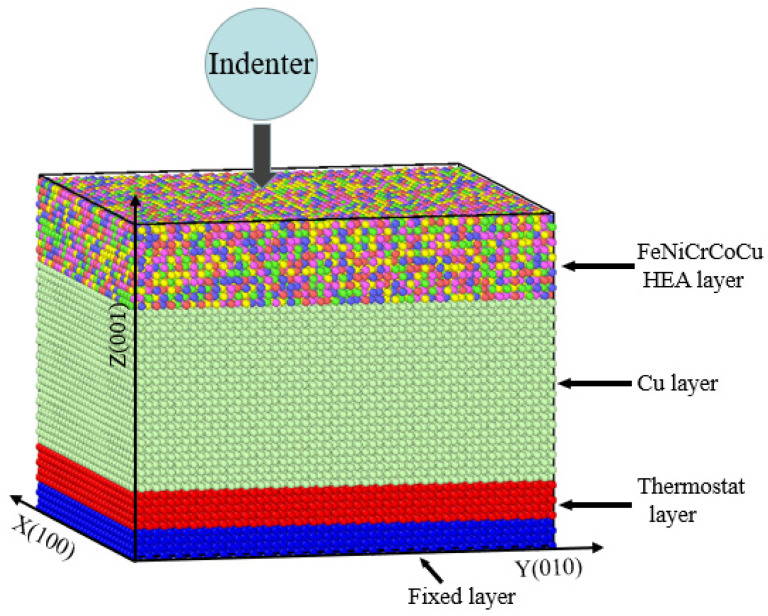
Schematic diagram of nanoindentation model area distribution.

**Figure 3 materials-17-03911-f003:**
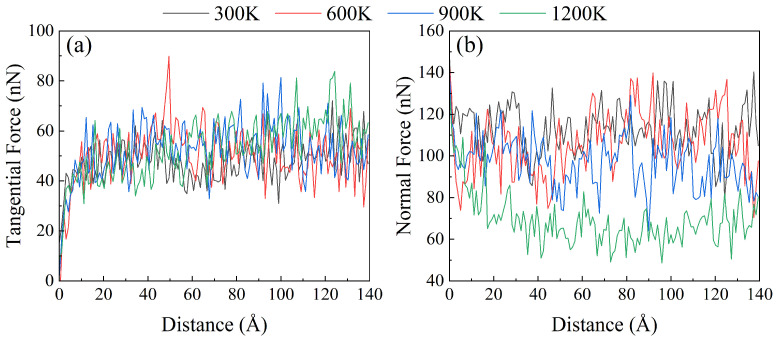
Variation curves of (**a**) tangential force and (**b**) normal force during friction under different temperatures.

**Figure 4 materials-17-03911-f004:**
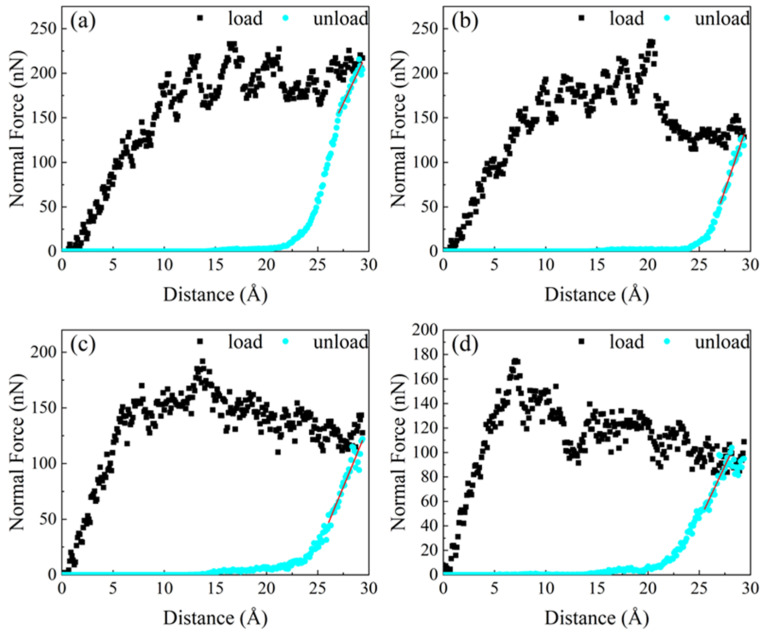
Nanoindentation loading curve at different temperatures of (**a**) 300 K, (**b**) 600 K, (**c**) 900 K, and (**d**) 1200 K.

**Figure 5 materials-17-03911-f005:**
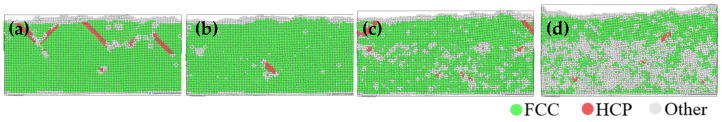
Lattice evolution after friction at different temperatures of (**a**) 300 K, (**b**) 600 K, (**c**) 900 K, and (**d**) 1200 K from the front view.

**Figure 6 materials-17-03911-f006:**
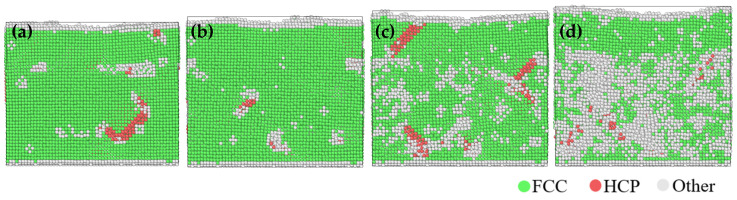
Lattice evolution after nanoindentation at different temperatures of (**a**) 300 K, (**b**) 600 K, (**c**) 900 K, and (**d**) 1200 K from the front view.

**Figure 7 materials-17-03911-f007:**
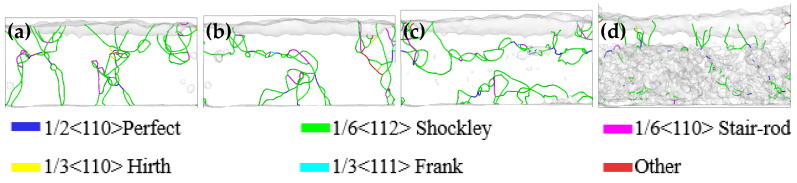
Dislocation evolution after friction at temperatures of (**a**) 300 K, (**b**) 600 K, (**c**) 900 K, and (**d**) 1200 K from the front view.

**Figure 8 materials-17-03911-f008:**
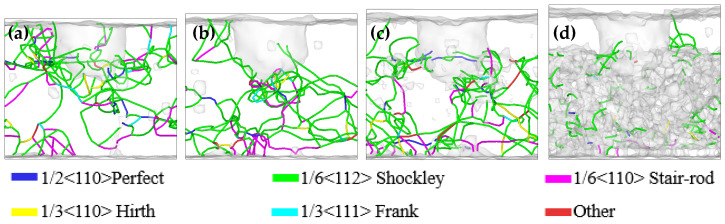
Dislocation evolution after nanoindentation at temperatures of (**a**) 300 K, (**b**) 600 K, (**c**) 900 K, and (**d**) 1200 K from the front view.

**Figure 9 materials-17-03911-f009:**
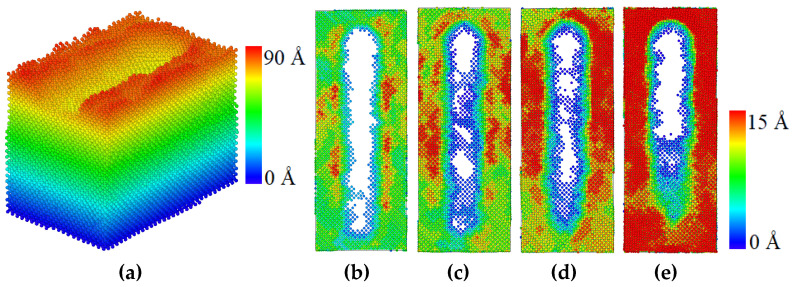
Atomic wear morphology: (**a**) three-dimensional morphology of atomic wear and atomic wear at temperatures of (**b**) 300 K, (**c**) 600 K, (**d**) 900 K, and (**e**) 1200 K from the top view.

**Figure 10 materials-17-03911-f010:**
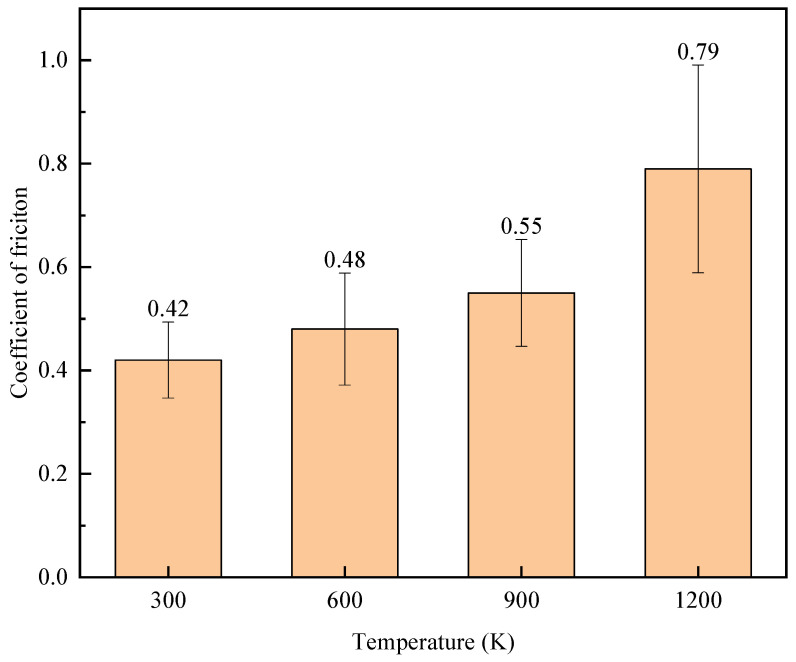
Friction coefficients of FeNiCrCoCu high entropy alloy coating material under different temperature.

**Figure 11 materials-17-03911-f011:**
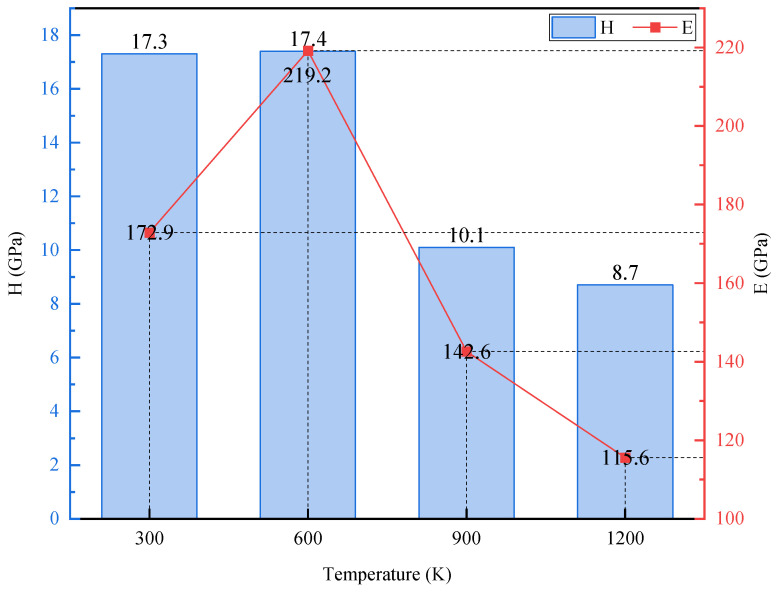
Hardness and elastic modulus under different temperature environments.

**Table 1 materials-17-03911-t001:** Relevant calculation parameters for elastic modulus and hardness.

Temperature/K	*h/*Å	*S*	*P*/nN	*v*
300	29.6	12.14	61.66	0.32
600	29.7	17.38	119.03	0.32
900	29.8	16.05	120.46	0.32
1200	29.7	19.93	122.43	0.32

## Data Availability

The original contributions presented in the study are included in the article, further inquiries can be directed to the corresponding author.

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
