# Peer review of "Molecular Dynamics Study on Wear Resistance of High Entropy Alloy Coatings Considering the Effect of Temperature"

_materials, 2024, doi:10.3390/ma17163911_

Round 1
Reviewer 1 Report
Comments and Suggestions for Authors
Dear Authors
You have a comprhensive work please tfor the first readers, your paper should be clear conclusions and aims and why you have chosen this materials etc.,
Author Response
Comments 1: You have a comprhensive work please for the first readers, your paper should be clear conclusions and aims and why you have chosen this materials etc. |
Response 1: Thank you for pointing these out. The aims of our paper are to analyze the influencing mechanism of wear resistance of FeNiCrCoCu high entropy alloy coatings at different temperatures from the microstructure. We have cleared these aims in the Introduction part –page 2, paragraph 3, and line 95-99. “Therefore, in response to the shortcomings of current research for high entropy alloys coatings, this study aims to analyze the influencing mechanism of wear resistance of FeNiCrCoCu high entropy alloy coatings on Cu matrix at different temperatures from the microstructure by molecular dynamics methods, to provide theoretical basis for practical applications.” The conclusions of our paper were more clearly restated in the Conclusions part –page 12, paragraph 1-3, and line 356-377. “In this study, the wear resistances of FeNiCrCoCu high entropy alloy coating material were comprehensively analyzed by molecular dynamics method. The effects of temperature on evolution of lattice structure, dislocation, friction coefficient and hardness of high entropy alloy coating material were analyzed. Through the above re-search and analysis, the following conclusions are obtained: (1) In the friction process of FeNiCrCoCu high entropy alloy coating, the value of the normal force decreases greatly due to thermal softening with the increase of temperature. Friction coefficients increased with the temperature, especially when temperature get 1200K. It is because that high temperature leads to the increase of the proportion of disordered atoms in the material. At 300K, 600K and 900K, the ordered lattice structures of the high entropy alloy coating material are basically the same. At the temperatures 1200K, the proportion of disordered lattice structure increases rapidly, resulting in a significant de-crease in normal force, which caused a significant decrease in friction coefficient. (2) When the temperature increased from 300K to 600K, the elastic modulus in-creased significantly. The density of dislocations at 600K is significantly reduced com-pared with that at 300K, resulting in the increase of the elastic modulus of the material from 173GPa to 219GPa. At the temperatures of 900K and 1200K, the densities of dislocations increase rapidly resulting in a significant decrease in the elastic modulus of the material. The hardnesses of FeNiCrCoCu high entropy alloy coating are basically the same at 300K and 600K. And the hardness keeps decreasing with increasing temperature, when the temperature reaches 600K. It could be found that the degree of atomic disorder in-creases with temperature when the temperature reaches 600K, which leads to decreasing in hardness.” The reasons why we have chosen this material was also further supplemented in the Introduction part – page 2, paragraph 2, line 53-59 and paragraph 3, line 89-93. “Among them, FeNiCrCoCu high entropy alloys has excellent wear resistance, which is regarded as ideal high-temperature wear-resistant materials, attracting a large number of scholars to study them. At present, it has been proven that this alloy can be achieved and is suitable for use as a thin film application [20-22]. Due to the expensive preparation of this high entropy alloy material, it is more reasonable to add it as a coating material to traditional metal surfaces in future practical applications.” “It can be concluded that researchers mainly focus on studying the room temperature friction performance of high entropy alloy materials and have not studied the overall wear resistance at different temperatures. However, at present, researchers mainly focus on the study of the properties of the materials themselves and have not studied their overall wear resistance as coating materials.” |

Reviewer 2 Report
Comments and Suggestions for Authors
Author Response
Comments 1: In the Materials and methods section, at page 3, rows 116 and 117 are mentioned two acronyms: NPT and NVT. In my opinion, I think it should be explained in parentheses what it means. |
Response 1: Thank you for pointing these out. We are sorry that the acronyms: NPT and NVT, were not mentioned in the paper. The explanations of these acronyms were added in detail–page 3, paragraph 2, and line 122-127. “We perform time integration on Nose-Hoover style non-Hamiltonian equations of motion which are designed to generate positions and velocities sampled from the canonical, isothermal-isobaric, ensembles. The canonical ensembles are collection of systems with the same number of particles and volume, and the temperature is fixed, but the energy is allowed to fluctuate (NVT). The isothermal-isobaric ensembles are systems with fixed number of particles, fixed pressure and fixed temperature (NPT).” |
Comments 2: Regarding figure 6, the explanations are missing. For example, why does the number of stacking faults increase as the temperature increases? Why does their number start to decrease during friction process? |
Response 2: Agree. We have accordingly modified the explanations of the results in original figure 6 (current figure 5). The corresponding explanations were stated – page 7, paragraph 1, line 208-211 and line 222-225. “It shows that the number of hexagonal close-packed (HCP) atoms increases after friction with the increase of temperature. The corresponding defect structures indicate new stacking faults were induced by friction, consistent with the previous observation that the planar slip mainly governs the deformation [44, 45].” “During the friction process, a large number of disordered structures appeared and began to melt, and the stacking faults also began to decrease at 1200K. Due to the softening of the lattice with increasing temperature, the deformation mechanism transformed from stacking faults to lattice disorder.” |
Comments 3: Also, regarding to figure 12, only some findings related to the elastic modulus are presented, with no reason explained. |
Response 3: Thank you for pointing these out. We have accordingly added the explanations for findings regarding to original figure 12 (current figure 11). The corresponding explanations were stated – page 3, paragraph 2, and line 122-127. “We perform time integration on Nose-Hoover style non-Hamiltonian equations of motion which are designed to generate positions and velocities sampled from the canonical, isothermal-isobaric, ensembles. The canonical ensembles are collection of systems with the same number of particles and volume, and the temperature is fixed, but the energy is allowed to fluctuate (NVT). The isothermal-isobaric ensembles are systems with fixed number of particles, fixed pressure and fixed temperature (NPT).” |

Reviewer 3 Report
Comments and Suggestions for Authors
This paper is devoted to systematically analyzing the wear resistance of FeNiCrCoCu high entropy alloy coating using the molecular dynamics method.
The topic of the research corresponds to the journal`s goals.
The paper is well structured, and the authors provided complex research.
The authors must add the paper`s goal and research tasks at the end of the Introduction. Also, please highlight the scientific novelty of this study.
The Figs. 1 and 2 are almost similar. I recommend combining them.
How can you verify the modeling results you obtained? What about physical experiments? Did you conduct them or plan to conduct them?
What are the main limitations of this research?
Please add future research plans to the conclusions.
Author Response
Comments 1: This paper is devoted to systematically analyzing the wear resistance of FeNiCrCoCu high entropy alloy coating using the molecular dynamics method. The topic of the research corresponds to the journal`s goals. The paper is well structured, and the authors provided complex research. |
Response 1: Thank you for the evaluation. We would strive to further improve the quality of the paper |
Comments 2: The authors must add the paper`s goal and research tasks at the end of the Introduction. Also, please highlight the scientific novelty of this study. |
Response 2: Agree. The goal and research tasks were added at the end of the Introduction and the scientific novelty of this study was highlighted– page 2, paragraph 3, line 89-99. “It can be concluded that researchers mainly focus on studying the room temperature friction performance of high entropy alloy materials and have not studied the overall wear resistance at different temperatures. However, at present, researchers mainly focus on the study of the properties of the materials themselves and have not studied their overall wear resistance as coating materials. And it is difficult to observe its microscopic deformation mechanism during the experimental process, and effective conclusions cannot be drawn on the impact mechanism of wear resistance at different temperatures. Therefore, in response to the shortcomings of current research for high entropy alloys coatings, this study aims to analyze the influencing mechanism of wear resistance of FeNiCrCoCu high entropy alloy coatings on Cu matrix at different temperatures from the microstructure by molecular dynamics methods, to provide theoretical basis for practical applications.” |
Comments 3: The Figs. 1 and 2 are almost similar. I recommend combining them. |
Response 3: Thank you for pointing this out. We have combined Figs. 1 and 2. |
Comments 4: How can you verify the modeling results you obtained? What about physical experiments? Did you conduct them or plan to conduct them? |
Response 4: We agree with this comment. We added experimental and simulation results of FeNiCrCoCu high entropy alloy research and compared them with the simulated data results in the article to ensure the reliability of the data. For example, the friction coefficient data was also basically consistent with 0.5~0.75 measured by Li et al. in the experiments [48]. Luo et al. simulated the FeNiCrCoCu high entropy alloy at 293 K and obtained an elastic modulus of 201GPa and a hardness of 17.2GPa, which is very close to the simulation results at 300K in this paper [51]. And the simulation results at 300K were also basically consistent with the results obtained by Deng et al. in the experiment [52]. We plan to conduct corresponding physical experiments at different temperature in subsequent research. The corresponding revisions were also highlighted in the re-submitted files–page 9, paragraph 2, line 308-309 and page 11, paragraph 3, line 343-353. “The friction coefficient data was also basically consistent with 0.5~0.75 measured by Li et al. in the experiments [48].” “Luo et al. [51] obtained the elastic modulus of 201GPA and hardness of 17.2GPA of FeNiCrCoCu high entropy alloy at 293K, which is very close to the simulation results at 300K in our work. And the results are also basically consistent with the results obtained by Deng et al. [52] in the experiment.” |
Comments 5: What are the main limitations of this research? Please add future research plans to the conclusions. |
Response 5: We have introduced the main limitations of this research at the conclusions part –page 12, paragraph 4, line 378-383. “There are still many shortcomings in this article. We didn’t analyze the effect of changes in the proportion of elements in FeNiCrCoCu high entropy alloy materials on wear resistance. Subsequent analysis can be conducted by changing the proportion of elements to obtain the external influencing mechanism. And we did not study the roughness of the material surface. In the future, the influence of roughness on the wear resistance of materials can be studied by changing the surface morphology of the material.” |
